# Modelling reveals the effect of climate and land use change on Madagascar's chameleons fauna
Alessandro Mondanaro[1], Mirko Di Febbraro[2], Silvia Castiglione[3], Arianna Morena Belfiore [2], Giorgia Girardi[3], Marina Melchionna[3], Carmela Serio[3], Antonella Esposito[3] & Pasquale Raia [3] ✉

The global biodiversity crisis is generated by the combined effects of human-induced climate change and land conversion. Madagascar is one of the World's most renewed hotspots of biodiversity. Yet, its rich variety of plant and animal species is threatened by deforestation and climate change. Predicting the future of Madagascar's chameleons, in particular, is complicated by their ecological rarity, making it hard to tell which factor is the most menacing to their survival. By applying an extension of the *ENphylo* species distribution model algorithm to work with extremely rare species, we find that Madagascar chameleons will face intense species loss in the north-western sector of the island. Land conversion by humans will drive most of the loss, and will intersect in a complex, nonlinear manner with climate change. We find that some 30% of the Madagascar's chameleons may lose in the future nearly all their habitats, critically jeopardizing their chance for survival.

The current biodiversity crisis is elicited by the combined effects of climate change and the other negative consequences deriving from human economic activities[1–3]. These effects are so penetrating that most scholars agree a sixth mass extinction is in fact underway[4,5]. The intensity of the ongoing extinction wave varies across taxonomic groups and geography, being ostensibly more severe among both the largest and smallest-bodied terrestrial vertebrates[4,6,7] and in tropical and subtropical regions[5,8]. With its extraordinarily diverse fauna exposed to intense human disturbance (mainly via land use change and deforestation) Madagascar stands out as one of the worst-hit regions in the world[9–12]. Predicting the future of biodiversity on this large island is complicated. The rapidly expanding network of protected areas may buffer extinction risk in unpredictable ways, depending on the availability of space and corridors left by humans to the species tracking their habitats in the wake of climate change[9,13,14]. The effects of climate change and land use conversion are not necessarily additive, as some species may even expand their habitats under the anticipated warmer temperatures, whereas the impact of land use conversion is mostly expected to be detrimental[15,16]. Moreover, producing solid bioclimatic models that can be projected to the future to assess where the habitats will be suitable for rare species is difficult[16–18]. The problem with modelling uncommon taxa is particularly sensitive in regions with high levels of endemism and narrow-ranged species like Madagascar[9,12,15]. Although a number of solutions were provided in the past to address the problem of rarity in modelling[19,20] at least 10 datapoints per species are usually required under such approaches, which

translates to geographic range sizes no less than 100 km$^2$ even with a fine-grained, $10 \times 10$ km gridded cell study design. This figure is larger, for instance, than the geographic range size of one fifth of Madagascar reptiles[21] and means that particularly rare species are usually excluded from studies addressing the future distribution of the biota (e.g. refs. 17,19,22–25). Unsurprisingly, the 10 datapoints limit was applied in modelling the expected future distribution of Madagascar's plants[16] and reptiles[17] and an even larger (20) datapoint limit was used to model typically narrow-ranged Madagascar's chameleons[23]. To solve this 'rare species modelling paradox'[26] we recently proposed a new algorithm, named *ENphylo*[27], which takes advantage of the strong phylogenetic signal in species climatic preferences[28] to infer the presumed climatic niche of rare species, and then provides habitat suitability models for them by intersecting the phylogenetically-imputed niche with the few observational datapoints available. *ENphylo* proved to be particularly accurate in modelling species for which as few as 10 to 20 observational datapoints where available, and outcompeted similar algorithms within this range[27]. Here, by using Madagascar's chameleons as a model, we expanded *ENphylo* to work with even rarer species, pushing modelling down to two observational datapoints only. The study system is ideal, at 95 taxonomically defined, mostly endemic species[21,23], Madagascar's chameleons are exceptionally diverse, represent a long and complex biogeographic history[9,29,30] and yet they are mostly infrequent, with some 30 species having range size smaller than 1000 km$^2$ [21]. This implies predicting their future distribution is problematic and further complicated by

[1]DST, University of Florence, 50121 Florence, Italy. [2]EnviXLab, Department of Biosciences and Territory, University of Molise, 86090 Pesche, (Isernia), Italy. [3]DiSTAR, University of Naples Federico II, 80126 Naples, Italy. ✉e-mail: pasquale.raia@unina.it

the uncertainty derived by the complex dynamics of Madagascar's landscapes and socioeconomic growth[9,29,31]. Albeit Madagascar's protected areas are expanding[13] and include at least in part the geographic range of some 97% of the threatened vertebrate species[9], the island is martyrised by decades of deforestation, which erased nearly one half of its forest cover[32] and still proceeds at a rate >80 kha/year[33]. In the 2000 to 2019 interval, the fraction of ecosystem services value generated by Madagascar forests decreased of some 3% while its total values increased by 7%, because of land use conversion to agriculture mainly at the expense of the forests[31], where most of the chameleon biodiversity occurrs[23,34]. To assess how the combined effects of future climate change and land use change on Madagascar's chameleon diversity, we modelled the potential future distribution of Madagascar's chameleons with *ENphylo*, including most of the rarest, under different scenarios of global warming and land use change, and tested how these factors will interact with each other in influencing their survival and distribution. We modelled the effects between climate and land use changes in terms of additive, agonistic or synergistic interactions, testing whether and where their expected contributions are expected to impinge on Madagascar's chameleon fauna.

## Results

We modelled 134 chamaeleonid species under different SDM algorithms. In particular, 25 out 56 Madagascar' chameleons were modelled under *ENphylo*. Fifteen out of 56 were modelled via ESM, and 16 more were eventually modelled by the model ensemble of MaxEnt, RF and GLM algorithms. Overall, we built 45 model predictions for each Chamaeleonidae species combining the three binarization thresholds, the three GCMs, and the two mild and severe scenarios for both climate and LULC, for a total of 1620 (45 maps × 56 species) predictive maps.

We assessed the effect of spatial autocorrelation by quantifying its amount in ensemble model residuals by means of Moran's I correlograms. According to this analysis, the average Moran's I value among all the species is equal to −0.13 (sd = 0.06), with only 14% of significant replicates. These results indicate an overall negligible effect of spatial autocorrelation on models.

Since evaluation metrics can provide misleading results when quantifying the accuracy of models calibrated on sample size <10 presences[35], we reported the *ENphylo* performances without considering the models related to the species with <10 occurrences. Under this filtering, *ENphylo* achieved fair-to-excellent predictive performances with an AUC value averaged among the modelled species equal to 0.918 (sd = 0.106), an average TSS equal to 0.703 (sd = 0.166), and an average Boyce index equal to 0.394 (sd = 0.122). It is worth nothing, though, that model selection (i.e., phylogeny selection) was implemented by selecting the replicate with the highest AUC value in *ENphylo*, meaning that TSS and Boyce values might be underestimated.

The other approaches led to equally robust model performances: averaged AUC = 0.953 (sd = 0.033), averaged TSS = 0.871 (sd = 0.086), and average Boyce index = 0.827 (sd = 0.067) for the ESMs, while averaged AUC = 0.964 (sd = 0.033), averaged TSS = 0.864 (sd = 0.100), and averaged Boyce index = 0.965 (sd = 0.04) for traditional SDMs.

Future projections showed large loss and gain areas for Chamaeleonidae species in different regions of Madagascar (Fig. 1, Supplementary Figs. 1–2). The magnitude of the species loss/gain estimates increased/decreased passing from the mild to severe future scenarios (Table 1). The impact of future LULC change for both loss and gain dynamics seems relevant at large spatial scale, whereas future climate predictions suggest an important role at local scale (Fig. 1, Supplementary Figs. 1–2). As for the interaction type, the "only LULC" effect has the highest percentage for both gain and loss dynamics, whereas the synergic effect has the lowest value. The role of antagonistic interaction is relevant for species loss only (Fig. 2, Table 2).

## Discussion

Madagascar is a hotspot of biodiversity, hosting a great variety of terrestrial tetrapods and plants unique to the island[12]. This rich and diversified biota is facing serious threats posed by humans via overexploitation of the natural landscape and species, and by human-driven climate change[9]. These activities are prolonging a pattern of extinction initiated by the first human colonizers who eradicated the island's megafauna[36–38]. Despite protected areas now cover 10.4% of the country and encompass, at least in part, the distributional range of virtually all threatened vertebrates[9] rapid deforestation has taken its toll on Madagascar biodiversity for almost fifty years now[32,33]. Besides direct human activities, climate change poses another obvious threat to the island's biota. Albeit climate change is anticipated to exert a negative effect on Madagascar's forested biomes[39], projected range maps suggest a mixed blessing, with most range contraction but also sizeable range expansion being forecasted[15,16,39]. Consequently, human disturbance is usually perceived as posing a greater threat to the survival of Madagascar's species compared to the potential effects of climate change[9]. Whereas vegetational cover has been readily modelled, though, the future distribution of the tetrapod fauna is less well-understood and new species are still being described[12], possibly because of the restricted geographic range of several, small-bodied endemic species[23]. Thus, the future of the small vertebrates is surrounded by greater uncertainty than with plants, which imperils the proper design of protected areas and corridors management to save them from extinction[13,14]. Here, we applied species distribution modelling to 56 Madagascar's chameleon species, including many of the rarest. The list of uncommon species comprise leaf chameleons *Brookesia* (*B. desperata*, *B. karchei*, *B. micra*, and *B. tristis*), several species in the genus *Calumma* (*C. guibei*, *C. furcifer*, *C. guillaumeti*, *C. marojezense*) and *Furcifer* (*F. timoni*, *F. bifidus*) whose range is mostly restricted to small areas in the North, like Montagne d'Ambre and Marojejy National Parks and Nosy Hara archipelago. Unfortunately, we found that for some 10–20 species the predicted geographic range change will exceed 90% loss, depending on the LULC and climate change future scenario (Supplementary Data 3), meaning they will be in grave danger of extinction. This set of imperilled taxa mostly occupy dry deciduous, subhumid and lowland forest habitats such as canopy chameleon *Furcifer willsii*, Decary's leaf chameleon *B. decaryi* and *B. brunoi*, flat-casqued chameleon *Calumma globifer*, and the abovementioned *B. desperata*, *B.karchei*, *B. micra*, and *B. tristis*, additional species within *Calumma* (*C. amber*, *C. guibei*, *C. ambreense*, *C. nasutum*, *C. fallax*, *C. peltierorum* and *C. boettgeri*) and *Furcifer* (*F. petteri*). In contrast, geographic range change predictions are positive (meaning they are expected to gain range) for widespread species, some of which occur in multiple habitats such as giant and rhinoceros chameleon *Furcifer oustaleti and F. rhinoceratus*, Parson's, blue-legged, short-horned and O'Shaughnessy's chameleons (*C. parsonii*, *C. crypticum*, *C. brevicorne*, and *C. oshaughnessyi*) and the stump-tailed chameleon *B. superciliaris*. Overall, though, there is no clear relationship between commonness and the predicted percentage of habitat loss, especially for the species occurring in the dry deciduous forest habitat, such as *Calumma ambreense*, *C. amber*, and *Brookesia tuberculata*, all predicted to be losing >90% of their habitat despite their commonness.

At comparing the effects of climate change and land use conversion, and their intersection, on species habitat suitability in the future, we found that the effects of LULC were preponderant. We predict that almost one chameleon species per cell in Madagascar will find unsuitable habitats in the future (Table 1). This result does not change by considering different land use future scenarios whereby the intensification of the climatic and socioeconomic factors will only affect the maximum of species loss ranging from 8 (mild) to 11 (severe) (Table 1, Supplementary Fig. 2). This is mostly due to changes in LULC, which are predicted to override the effects of climate change and the interaction terms of both sets of variables, regardless of which climate change scenario, or SDM parameters are used (Fig. 2, Table 2). The highest chameleon biodiversity loss is expected to occur in the dry deciduous forest in the western and north-western sectors of the island (Fig. 1, Supplementary Fig.1 and Supplementary Data 4). In contrast, most potential species losses and gains (hence higher turnover) are concentrated in the lowland forest in the eastern sector of the island (Fig.1). The true nature of species gains, though, is hard to tell. Chameleons are not particularly good at dispersal, and we consequently bounded dispersal distance to

**Fig. 1 | Species loss and gain of the Chamaeleonidae species future projected distributions under the severe SSP scenario.** Top: Species loss and gain calculated considering the dynamic climate ("ssp585_lc_recent"), dynamic land use ("lc_85_dinam"), and the dynamic land-climate ("ssp585_lc85"). Predictions were obtained by averaging the results derived from all thresholds and GCM combinations (bottom).

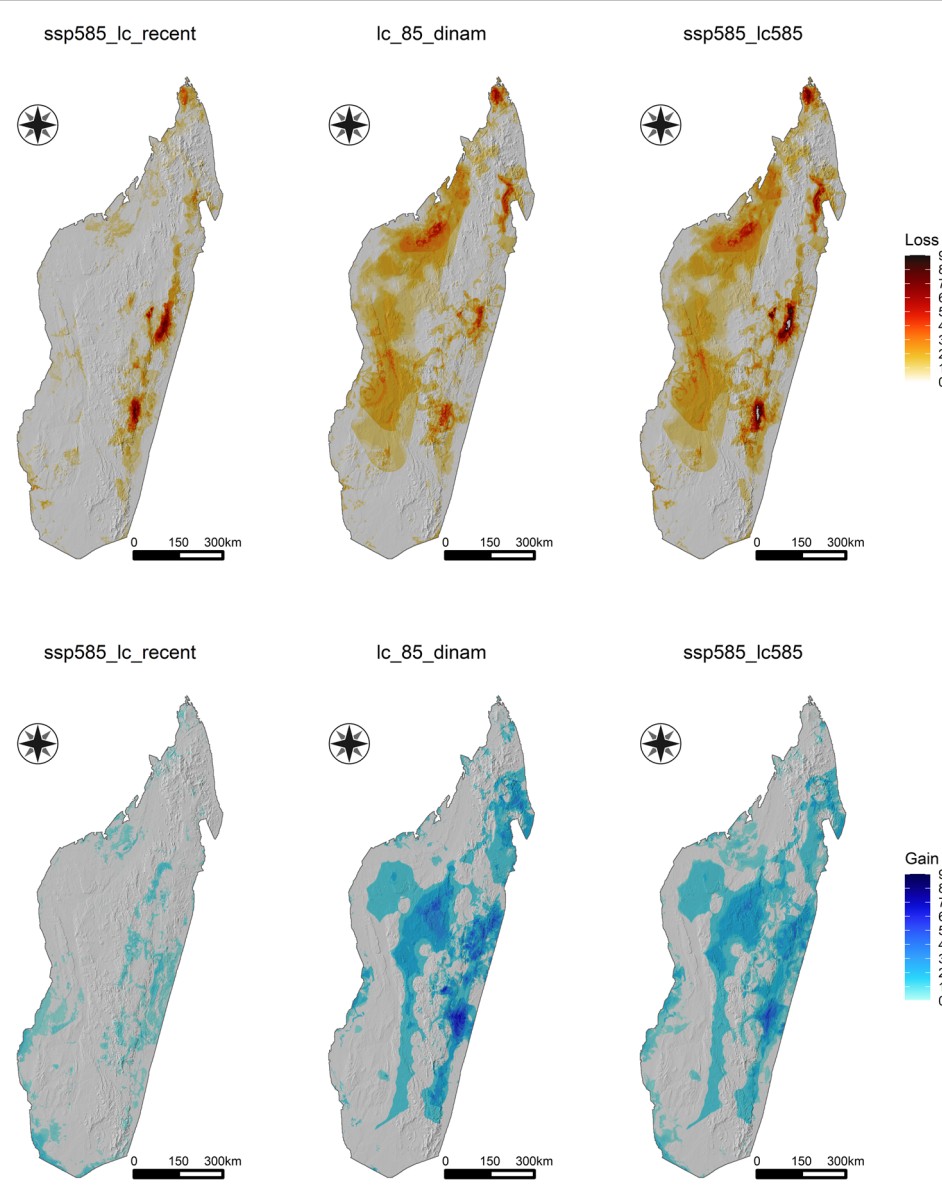

around 1 km yr$^{-1}$ [40]. While this figure is not particularly high by vertebrate standards, it is liberal in assuming the habitat corridors can in fact be trespassed. This will critically depend on their existence and functioning and ignores lack of connectivity between suitable habitats. This implies the potential gains, in the absence of human-mediated translocation, are hypothetical at best.

Predicting species distribution for extremely rare species cannot usually be done[26,41], and they are usually omitted[18] or presence datapoints simulated within the extent of occurrence so that up to 10 geographic cells are included[35]. With SDM validation, statistical issues introduced by rarity are still present, as with rare species common validation metrics such the AUC suffers from variation in the species prevalence and sample size[42]. In addition, pseudo-presences necessarily introduce some autocorrelation in the data, and can only be supposed to may harbour the species presence. However, it is important to notice that our procedure is not meant to 'learn' the niche from the 'pseudo-presences'. In *ENphylo*, the niche of a rare species is imputed after modelling the niches of its relatives with ENFA. True presences and 'pseudo-presences' are just used to convert the imputed niche into habitat suitability, by computing the Mahalanobis distance between sampled and the 'mean' habitat[43,44]. Since the 'pseudo-presences' are sampled exactly according to their similarity to the real presence datapoints, this

distance could be safely assumed to be left almost unaltered by our procedure. Of course, this does not mean that five datapoints provide a faithful representation of the species niche, especially because we selected the pseudo-presences as to minimize their dissimilarity from the true presences. This may possibly imply the rare species niches are probably narrow. We believe maintaining this non-random narrowness, though, provides a more genuine representation of the actual niche than producing an artificially broad niche for an otherwise little-known species. As a matter of fact, *ENphylo* predictive performance was quite high (AUC > 0.9) which testifies that the modelling procedure was robust. By applying *ENphylo*, and by testing explicitly the combined effects of land-use and climate change under different scenarios, we found that the most serious threat to Madagascar's future chameleon biodiversity is posed by land conversion to agriculture and other human activities, rather than by climate change.

## Materials and methods
### Chamaeleonidae occurrences
We downloaded the Chamaeleonidae modern occurrences from the "Global Biodiversity Information Facility" online database (GBIF; www. gbif.org/), including only the data provided with geographical coordinates. Data were further filtered by selecting "Material citation and both Machine

**Table 1 | Species loss and gain of the Chamaeleonidae species**

| | Average loss | | Average gain | | Max loss | | Max gain | |
|---|---|---|---|---|---|---|---|---|
| | Mild | Severe | Mild | Severe | Mild | Severe | Mild | Severe |
| **Dynamic climate** | 0.163 | 0.275 | 0.060 | 0.067 | 6.000 | 8.889 | 2.444 | 2.000 |
| **Dynamic land use** | 0.749 | 0.737 | 0.722 | 0.573 | 6.667 | 7.667 | 8.000 | 8.000 |
| **Dynamic land-climate** | 0.823 | 0.890 | 0.677 | 0.470 | 8.889 | 11.222 | 6.889 | 6.111 |

Species loss and gain calculated under both the mild (SSP1-2.6) and severe (SSP5-8.5) future predictions. Loss and gain values were obtained by averaging the predictions derived from all thresholds and GCM combinations. For each scenario, the average and maximum loss and gain values are reported.

**Fig. 2 | Interaction effects of climate and land use change on Chamaeleonidae gain and loss.** Top: Species loss and gain calculated considering the severe SSP, MaxSens+Spec as threshold, and MRI-ESM2-0 as CMIP6 scenario (bottom).

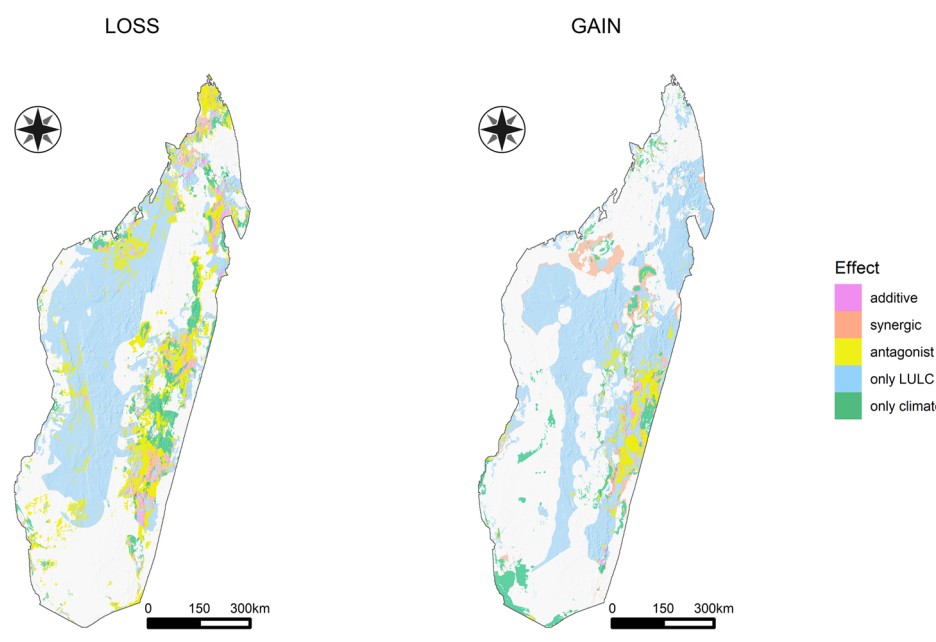

and Human observations" as the basis of record categories (GBIF.org (26 March 2024) GBIF Occurrence Download https://doi.org/10.15468/dl.86bqsw). Record accuracy was assessed by including only occurrences given to at least two decimal places (0.01 decimal degrees, corresponding to 1.11 km at the equator) and by removing duplicated or unrealistic records. Then, we excluded the data outside the African continent and the Mediterranean area. Overall, we gathered 17170 occurrences belonging to 151 Chamaeleonidae species.

### Environmental variables

To address the potential impact of climate and land use change on the future of Madagascar's chameleons, we started by considering the 19 bioclimatic variables listed in the CHELSA database version 2.1[45] (http://chelsa-climate.org/) as environmental predictors. Specifically, we downloaded the high-resolution (1 × 1 km) modern climatic data (1981–2010) as the reference temporal period, while for future scenarios, we referred to projections for the 2071–2100 interval. We took into account both the mild and severe "Shared Socioeconomic Pathways" (SSP), i.e., SSP1-2.6 and SSP5-8.5 scenarios. For each of them, we considered three global circulation models (GCMs) from the Coupled Model Intercomparison Project (CMIP6), namely GFDL-ESM4, MRI-ESM2-0, and IPSL-CM6A-LR. To take into account Madagascar's land use change scenarios, we considered seven Land Use/Land Cover (LULC) categories (https://www.geosimulation.cn/), as provided in ref. 46. Specifically, we calculated Euclidean distances from each of the seven LULC categories and used them as predictors. Climate and LULC variables were rasterized at 1 km spatial resolution and cropped along Madagascar extent. Subsequently, the variables were checked for multicollinearity by using the "usdm" R package[47]. After excluding the variables with a high Pearson's correlation coefficient (using 0.7 as the threshold), we retained 11 predictors: Temperature Seasonality (BIO4), Max Temperature of Warmest Month (BIO5), Temperature Annual Range (BIO7), Precipitation of Wettest Month (BIO13), Precipitation Seasonality (BIO15) and Euclidean distance from Water bodies, Forests, Grasslands, Barren areas, Urban areas, and Croplands.

### Species distribution models (SDMs)

The selected predictors were used to feed species distribution models (SDMs) sought to predict the current and future potential distribution of Chamaeleonidae species. We used three different modelling approaches depending on the number of occurrences available per species. Specifically, we adopted the *ENphylo* modelling algorithm[27] for the species with less than 15 occurrences, given the proven ability of this algorithm to outperform ensembles of small models (ESMs)[20] and "traditional" SDMs when rare species are modelled[27]. To run *ENphylo*, for each species we randomly generated 10,000 background points across the ecoregion included within the study area[48]. In *ENphylo*, the climatic niche dimensions of rare species are derived estimating the marginality and specialization axes of well-sampled species under Ecological-Niche Factor Analysis (ENFA)[44] analysis first, and then estimating marginality and specialization for the rare species via phylogenetic imputation. Although this procedure provides the essential niche information, at least five occurrences are necessary to convert the phylogenetically-imputed niche marginality and specialization axes into Mahalanobis distances from the available climates and then into habitat suitability values[27]. Here, we expanded *ENphylo* to work with less than five occurrences. To this aim, we started designing as pseudo-presences the cells adjacent to true presence ('reference') cells, according to the knight move

**Table 2 | Percentages of the interaction effects of the possible combinations of climate and land use change across the study area**

| | | Syngergic | Additive | Only climate | Antagonistic | Only LULC |
|---|---|---|---|---|---|---|
| **Mild scenario** | Loss | 1.70% | 2.70% | 5.28% | 15.13% | 75.19% |
| | Gain | 3.62% | 0.86% | 6.57% | 3.68% | 85.27% |
| | | syngergic | additive | only climate | antagonistic | only LULC |
| **Severe scenario** | Loss | 1.59% | 4.04% | 8.16% | 19.16% | 67.06% |
| | Gain | 6.21% | 0.92% | 9.51% | 4.62% | 78.74% |

The percentages are averaged overall threshold and GCM combinations under mild (SSP1-2.6) and severe (SSP5-8.5) future scenarios.

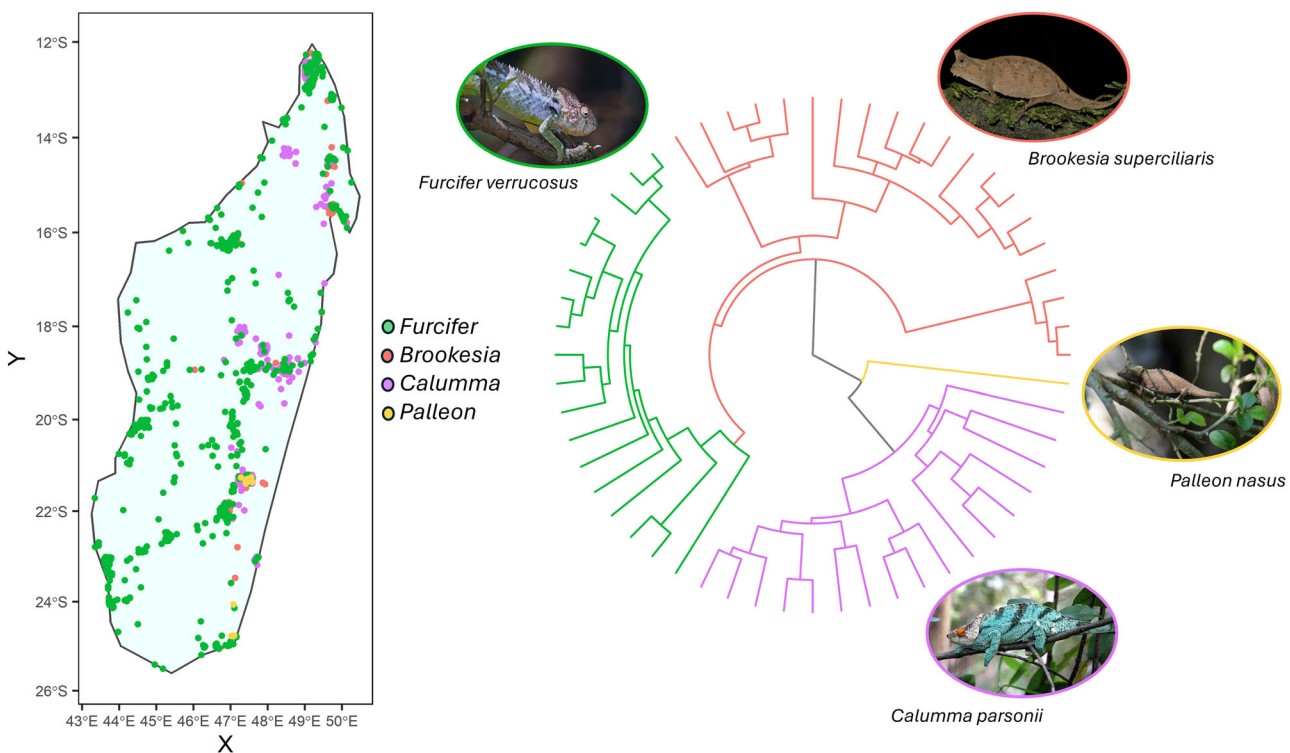

**Fig. 3 | Map of the occurrences of the chameleon data used to perform this study and their phylogenetic relationships.** (Top) The colours and tree refer to chameleons living in Madagascar. (Bottom) The image of *Furcifer verrucosus* is distributed under CC BY-SA 2.0 https://commons.wikimedia.org/wiki/File:Warty_Chameleon_(Furcifer_verrucosus)_(9628372559).jpg. The image of *Brookesia superciliaris* is distributed under CC0. https://commons.wikimedia.org/wiki/File: Brookesia_superciliaris_185939354.jpg. The image of *Palleon nasus* is distributed under CC BY-SA 4.0 https://commons.wikimedia.org/wiki/File:Palleon_nasus.JPG. The image of *Calumma parsonii* is distributed under CC BY-SA 4.0. https://upload.wikimedia.org/wikipedia/commons/4/42/Calumma_Parsonii_Ste_Marie_Madagascar.jpg.

criterion[49]. Then, we reduced the number of pseudo-presences by selecting among them those closest to the reference cells in terms of climate and LULC values. We first quantified the angle (i.e. the correlation coefficient) between the climatic/LULC vectors associated to reference cells and those of the pseudo-presence cells by computing the angle between their respective climatic/LULC vectors using the R package "RRphylo"[50]. Then, we selected $n$ pseudo-presence cells, where $n$ is equal to 5 minus the number of reference cells, according to their climatic/LULC similarity to the reference cells. This procedure is meant to select a minimum number of pseudo-presences as to attain to 5 potential presence cells (including the references) that could be used to convert the phylogenetically-imputed niche marginality and specialization into habitat suitability values[27]. To perform phylogenetic imputation of the niche marginality and specialization axes, we assembled a composite, informal supertree using the chameleon phylogeny in Tonini et al. [51] and Giles et al. [34] using the function *tree.merger*[52] in "RRphylo". We excluded species for which (i) geographical information was not available, (ii) occupy a single geographic cell, or (iii) phylogenetic information was not

available or conflicting between the source trees. The resulting phylogeny includes 134 species (56 of them currently living in Madagascar, Supplementary Data 1). After removing duplicate occurrences per cell, the total occurrence number amounts to 6915 (Fig. 3, Supplementary Data 2).

*ENphylo* models were assessed through a random bootstrap cross-validation procedure with replacement method by splitting the data into 80–20% training/testing folds. The bootstrap was repeated 10 times. Model predictive accuracy was assessed by calculating the area under the receiver operating characteristic curve (AUC), True Skill Statistic (TSS[53]), and Boyce index[54] and removing the models with an AUC value < 0.7. To account for phylogenetic uncertainty, the entire procedure was repeated by testing 50 alternative phylogenies, produced randomly modifying the species topology and branch lengths with the function *swapOne* in the R package "RRphylo"[50]. Then, we selected the model with the best AUC value over the 50 replicates. Since we were interested in species diversity changes in Madagascar, we retrieved only the best-fit models associated with Chamaeleonidae species which currently live in Madagascar and projected them

on current Madagascar climate/LULC as well as on future scenarios. Specifically, we considered the following climate and LULC change scenarios: (i) dynamic climate while LULC was held constant (hereafter "dynamic climate"), (ii) dynamic LULC while climate was held constant (hereafter "dynamic land use"), and (iii) dynamic LULC and dynamic climate (hereafter dynamic "LULC-climate"), following the same approach described in ref. 55. Both current and future model predictions were binarized to obtain presence/absence maps by using three thresholding schemes, as to account for the effect of adopting different binarization approaches[56]. Specifically, we selected the 'equalize sensitivity and specificity' (SensSpec), 'maximize TSS' (MaxSens+Spec), and 'minimum training presence' (TenPerc) using the "PresenceAbsence" R package[57].

To model species climatic niche, we adopted the ESM approach for species reporting a number of occurrences between 15 and 30. ESMs were trained by considering all possible combinations of the environmental variables taken two at a time. Lastly, for species with >30 occurrences were modelled by applying a traditional SDM approach (i.e. including all environmental variables at the same time). For both ESMs and SDMs, we adopted an ensemble forecasting approach by testing three widely used modelling techniques: Maximum Entropy (MaxEnt[58]), Random Forest (RF[59]), and Generalized Linear Models (GLM). Models were trained by relying on the functionalities provided in the 'biomod2' R package[60]. Specifically, we set "quadratic" and interaction level = 1 for defining the GLM parameters. For the other algorithms, we maintained the default parameter settings adopted in biomod2.

To evaluate model predictive accuracy, we performed a random bootstrap cross-validation with replacement scheme splitting the data into 80–20% training/testing samples and repeating this procedure 10 times. Model accuracy was evaluated by measuring the AUC, TSS, and Boyce index. Poorly calibrated models were avoided removing those with an AUC value < 0.7. Model averaging was performed by weighting the individual model projections by their AUC values and averaging the results[61]. Models were projected on the current Madagascar climate/LULC and on the future scenarios described above. Model projections were subsequently binarized by using the same thresholds described before.

Since the Chamaeleonidae species are characterized by limited dispersal abilities and live in highly-fragmented habitats[62], we decided to incorporate a dispersal constraint to the future projections of their distribution. We established a distance equal to 1 km as maximum annual dispersal rate, maintaining constant this value over time for all species. According to this strategy, we first calculated the minimum convex polygon enclosing all the localities where a species occurs, then created a buffer around this polygon with a radius equal to the maximum reachable distance from nowadays to the future (i.e., 60 km over a time span from 2010 to 2070). Subsequently, the current and future binary maps were cropped accordingly.

Binary maps obtained from the three model approaches were stacked among the species for current and future scenarios separately. After that, we calculated three indices: species richness (SR), species loss (L), and species gain (G), as to obtain a single vector of three values at each grid cell following the approach described in ref. 63. All indices were calculated by using the "biomod2".

To quantify the individual and synergic effects of the climatic and LULC changes in predicting Chamaeleonidae distribution, we calculated the delta in species richness (i.e., future − current) and summed the species losses and gains per cell by applying a paired comparison between the current and all the future scenarios. Moreover, the loss and gain values obtained for the dynamic land use (a), dynamic climate (b) and dynamic land-climate future scenarios (c) were adopted to define five types of interactions among the scenarios, as follows: (i) "Synergistic", when $c > a + b$, (ii) "Additive", when $c = a + b$, and (iii) "Antagonistic", when $c < \min(a, b)$ or $c < \max(a, b)$, or $\max(a, b) \leq c < a + b45$, (iv) "only climate" when $a = 0$ and $b > 0$, and "only LULC" when $a > 0$ and $b = 0$. All the procedure was repeated for loss and gain separately after removing the cells where no interaction occurred (i.e., all a, b, and c = 0).

Lastly, we quantified the types of interactions separately for mild and severe SPP under different future scenarios calculating the percentage of grid cells where an antagonistic, synergistic, and additive effect occurred.

The software developed to produce the rare species modelling, including ENphylo, is embedded in the R package RRdtn[64].

## Statistics and reproducibility
R scripts associated with this study are available as Supplementary Data 5.zip file. Files contains three annotated R scripts which describe the entire procedure used to model the Chamaeleonidae species by using the three different modelling approaches (ENphylo, ESM, and SDM). Codes are annotated to allow reproducibility.

## Reporting summary
Further information on research design is available in the Nature Portfolio Reporting Summary linked to this article.

## Data availability
Raw data and the chameleon phylogenetic tree are available in the Supplementary Data files.

## Code availability
R scripts associated with this study are available as Supplementary Data 5.zip file. The R package RRdtn embedding the ENphylo functions is available via Zenodo at https://doi.org/10.5281/zenodo.12734585.

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

## Acknowledgements
We are grateful to Francesco Carotenuto for his precious advice given in commenting some of the statistical approaches developed here.

## Author contributions
AM, MDF and PR conceived the study. AMB, GG, MM, CS and AE collected the data and helped with the preparation of the manuscript. AM conducted the main analyses with inputs by SC and MDF. PR and AM wrote the first draft with significant inputs by all the authors.

## Competing interests
The authors declare no competing interests
