## [Peer Review File · Communications Biology]

Reviewers' comments:

Reviewer #1 (Remarks to the Author):

In the paper "Modelling the rarest. The effect of climate and land use change on Madagascar's chameleons fauna" the authors modelled the potential future distribution of Madagascar's chameleons with ENphylo, including most of the rarest, under different scenarios of global warming and land use change, and tested how these factors will interact with each other in influencing their survival and distribution. This manuscript is well organized, and the drawn conclusions are coherent with the obtained results. I hope to provide very useful suggestions to improve the overall clarity of your study as well as the quality of your analysis. I think that my suggestions look feasible to you, and I believe you will be able to address them. Please see below.

Lines 85 – 86: Could you elaborate on your forecasts and hypotheses? If you want to be more specific about what you intend to do, you should enlarge this area.

Lines 100 – 101: Did you analyse your data for spatial autocorrelation?

Lines 104: Please include the specific link to the site from where you downloaded the data for each variable you utilized in your study (e.g., CHELSA and LULC).

Line 115: Please add all the R code (well commented!) used in your analysis in txt/word files stored in your supplementary materials (e.g., ENphylo and PresenceAbsence code R package).

Lines 261 – 335: I think that the authors should be discussing their results and comparing them with those already published on other species, genera, and families. Actually, your research addresses discoveries in regard to part of the field's work, but it leaves out other significant work that, in my opinion, should be included in your discussion when taking other niche analysis aspects into account.

Reviewer #2 (Remarks to the Author):

The authors studied the impact of environmental changes on chameleons using ENphylo for rare species, an ensemble of small models, and regular niche modeling. I think the authors did a pretty good job. However, I have several concerns regarding the reliability of the evaluation of their models, as their species have a low number of occurrences. They should also discuss more about the assumptions of their models regarding their species. Here are my detailed comments. I hope I can help the authors improve their manuscript.

L115-120: Please explain why you kept these specific predictors. How are they important regarding your taxa?

L156-157 & 186: Is it a bootstrap (thus selection with replacement) or a repeated split-sample cross-validation (without replacement)? Please make it clearer. In addition, how can you correctly evaluate the models with species with such a low number of occurrences (only one occurrence with a number of presences of 5)? Please read Collart & Guisan, 2023 (<https://doi.org/10.1016/j.ecoinf.2023.102106>). Finally, as you are working with presence-only data, I recommend evaluating your models also with the Boyce Index, which was designed for this type of dataset.

L183-185: Which R packages did you use and, more importantly, what were your model parameters? In your discussion, I suggest you discuss the assumptions/limitations of your models, especially regarding your rarest species. Do you think that with fewer than 5 occurrences you can capture the whole niche of the species (see van Proosdij et al., 2016 [<https://doi.org/10.1111/ecog.01509>] or Erickson & Smith, 2023 [<https://doi.org/10.1111/ecog.06500>])? You assume that all your species will conserve their niche in the future (especially for ENphylo, where all the links should stay the same). Do you know if local adaptations are possible regarding your species?

point-by-point replies to reviewers

Reviewers' comments:

Reviewer #1 (Remarks to the Author):

In the paper “Modelling the rarest. The effect of climate and land use change on Madagascar’s chameleons fauna” the authors modelled the potential future distribution of Madagascar’s chameleons with ENphylo, including most of the rarest, under different scenarios of global warming and land use change, and tested how these factors will interact with each other in influencing their survival and distribution. This manuscript is well organized, and the drawn conclusions are coherent with the obtained results. I hope to provide very useful suggestions to improve the overall clarity of your study as well as the quality of your analysis. I think that my suggestions look feasible to you, and I believe you will be able to address them. Please see below.

RE: Thank you for your kind words

Lines 85 – 86: Could you elaborate on your forecasts and hypotheses? If you want to be more specific about what you intend to do, you should enlarge this area.

RE: Thanks for suggesting. We added details about what we plan to do in the manuscript.

Lines 100 – 101: Did you analyse your data for spatial autocorrelation?

RE: Thanks for suggesting. We tested for autocorrelation in the models’ residuals and found comforting evidence that autocorrelation is not an issue. The text now reads:

“We assessed the effect of spatial autocorrelation by quantifying its amount in ensemble model residuals by means of Moran’s I correlograms. According to this analysis, the average Moran’s I value among all the species is equal to -0.13 (sd = 0.06), with only 14% of significant replicates. These results indicate an overall negligible effect of spatial autocorrelation on models.”

Lines 104: Please include the specific link to the site from where you downloaded the data for each variable you utilized in your study (e.g., CHELSA and LULC).

RE: We added the links to both CHELSA and LULC repositories in the new version of paper. Thanks for suggesting.

Line 115: Please add all the R code (well commented!) used in your analysis in txt/word files stored in your supplementary materials (e.g., ENphylo and PresenceAbsence code R package).

RE: As per the reviewer’s suggestion, we uploaded as Supplementary Files four annotated R scripts, which describe the entire workflow of all the analyses step by step. Specifically, the files “ENphylo_workflow.R”, “SDM_workflow.R”, and “ESM_workflow.R” show the procedure to run the three different modelling approaches. One additional file, “Species_diversity_workflow.R”, describes all the

procedure to calculate the species gain/loss and richness change under each future scenario. Additionally, the script illustrates how we averaged the results accounting for all threshold and GCM combinations.

Lines 261 – 335: I think that the authors should be discussing their results and comparing them with those already published on other species, genera, and families. Actually, your research addresses discoveries in regard to part of the field's work, but it leaves out other significant work that, in my opinion, should be included in your discussion when taking other niche analysis aspects into account.

RE: We wish there were more information. The Madagascar's chameleons distribution is so badly known (for many species) that, for instance, there is scanty information for 15 of 18 *Furcifer* species (arguably one of the best-known genus, <https://www.conservationleadershipprogramme.org/project/conservation-framework-furcifer-chameleons-madagascar/>). In his famous 2003 paper, Raxworthy and associates used museum records to know about the current distribution of the chameleons exactly because the information was so scarce. In highlighting the extraordinary biodiversity of Madagascar, Myers et al. (2000) cited Madagascar as one of the eight "hottest hotspots", but never mentioned chameleons. Similarly, in Ralimanana et al. 2022 Science paper, aptly titled "Madagascar's extraordinary biodiversity: Threats and opportunities" chameleons are never cited. In practice, either we go species by species for 72 chameleons (which clearly makes little sense and is impractical in terms of space) or no general comparison is, we feel, documented enough to highlight any major difference with the standing knowledge.

Literature cited

Raxworthy, C., Martinez-Meyer, E., Horning, N. et al. Predicting distributions of known and unknown reptile species in Madagascar. *Nature* 426, 837–841 (2003). <https://doi.org/10.1038/nature02205>

Myers, N., Mittermeier, R., Mittermeier, C. et al. Biodiversity hotspots for conservation priorities. *Nature* 403, 853–858 (2000).

Hélène Ralimanana et al., Madagascar's extraordinary biodiversity: Threats and opportunities. *Science* 378, eadf1466 (2022). DOI:10.1126/science.adf1466

Reviewer #2 (Remarks to the Author):

The authors studied the impact of environmental changes on chameleons using ENphylo for rare species, an ensemble of small models, and regular niche modeling. I think the authors did a pretty good job. However, I have several concerns regarding the reliability of the evaluation of their models, as their species have a low number of occurrences. They should also discuss more about the assumptions of their models regarding their species. Here are my detailed comments. I hope I can help the authors improve their manuscript.

L115-120: Please explain why you kept these specific predictors. How are they important regarding your taxa?

RE: It has been repeatedly demonstrated that chameleon species are highly vulnerable to combined effect of climate change and habitat transformation (Clark et al. 2024; Houniet et al. 2009). Madagascar fauna experienced massive environmental changes associated with both climatic conditions and human presence since the Late Holocene (Virah-Sawmy et al. 2010; Burney et al. 2004). It is also recognized as one of the ecoregions with the highest number of expected vertebrate species extinctions due to habitat loss linked to deforestation (Gonçalves-Souza et al. 2020). Consequently, we are confident that climatic and land use variables are the most important predictors for determining the future survival of chameleon species in Madagascar island. We specified this reasoning in the current version of the manuscript.

Literature cited

Clark, T. K., Alexander, G. J., & Tolley, K. A. (2024). Susceptibility of dwarf chameleons to climate and land use change: a vulnerability framework for conservation planning. *African Zoology*, 59(1), 26-38.

Houniet, D. T., Thuiller, W., & Tolley, K. A. (2009). Potential effects of predicted climate change on the endemic South African Dwarf Chameleons, *Bradypodion*. *African Journal of Herpetology*, 58(1), 28-35.

Virah-Sawmy M, Willis KJ and Gillson L (2010) Evidence for drought and forest declines during the recent megafaunal extinctions in Madagascar. *Journal of Biogeography* 37(3): 506–519.

Burney, D.A. et al. (2004) A chronology for late prehistoric Madagascar. *J. Hum. Evol.* 47, 25–63

Gonçalves-Souza, D., Verburg, P. H., & Dobrovolski, R. (2020). Habitat loss, extinction predictability and conservation efforts in the terrestrial ecoregions. *Biological Conservation*, 246, 108579.

L156-157 & 186: Is it a bootstrap (thus selection with replacement) or a repeated split-sample cross-validation (without replacement)? Please make it clearer.

RE: To validate the models, we adopted a random bootstrap cross-validation procedure with replacement method (Hastie et al. 2009). For both SDM and ESM, we chose “random” as cross-validation strategy by using the functionalities of biomod2 R package (Thuiller et al. 2009). We clarified the approach in the main text.

Literature cited

Hastie, T., Tibshirani, R., Friedman, J., Hastie, T., Tibshirani, R., & Friedman, J. (2009). Model inference and averaging. *The elements of statistical learning: Data mining, inference, and prediction*, 261-294.

Thuiller, W., Lafourcade, B., Engler, R. & Araújo, M. B. BIOMOD - A platform for ensemble forecasting of species distributions. *Ecography* 32, 369–373 (2009).

In addition, how can you correctly evaluate the models with species with such a low number of occurrences (only one occurrence with a number of presences of 5)? Please read Collart & Guisan, 2023 (<https://doi.org/10.1016/j.ecoinf.2023.102106>).

RE: The reviewer is correct by arguing that model evaluation is highly dependent on sample size. Model evaluation metrics can provide misleading results when quantifying the accuracy of models calibrated on sample size lower than 10 presences (Jiménez-Valverde, 2020). However, we demonstrated that ENphylo outperformed both SDM and ESM when rare species are modelled (Mondanaro et al. 2023). Specifically, we adopted AUC, TSS and Boyce index only for comparison between different SDM approaches because ENphylo predictions are completely free from the starting sample size. Indeed, the marginality and specialization matrices used to reconstruct habitat suitability values are phylogenetically imputed without relying on presence datapoints. Anyway, we calculated the AUC, TSS and Boyce index only for the species with at least 10 occurrences, further clarifying our approach in the main text as follows:

“Since evaluation metrics can provide misleading results when quantifying the accuracy of models calibrated on sample size lower than 10 presences⁵⁵, we reported the ENphylo performances without considering the models related to the species with less than 10 occurrences. After this filtering, ENphylo achieved fair-to-excellent predictive performances with an AUC value averaged among the modelled species equal to 0.918 (sd = 0.106), an average TSS equal to 0.703 (sd = 0.166), and an average Boyce index equal to 0.394 (sd = 0.122). It is worth nothing, though, that model selection (i.e., phylogeny selection) was implemented by selecting the replicate with the highest AUC value in ENphylo, meaning that TSS and Boyce values might be underestimated.”

Literature cited

Mondanaro, A., Di Febbraro, M., Castiglione, S., Melchionna, M., Serio, C., Girardi, G., ... & Raia, P. (2023). ENphylo: A new method to model the distribution of extremely rare species. *Methods in Ecology and Evolution*, 14(3), 911-922.

Jiménez-Valverde, A. Sample size for the evaluation of presence-absence models. *Ecol Indic* 114, 106289 (2020). [THIS IS REF 55]

Finally, as you are working with presence-only data, I recommend evaluating your models also with the Boyce Index, which was designed for this type of dataset.

RE: The reviewer is certainly correct arguing that Boyce index is recommended for evaluating models calibrated with presence-only data. Nonetheless, for the calculation of the continuous Boyce index, the habitat suitabilities were first ordered from lowest to highest values. Then, a moving window was used to calculate the ratio of the predicted frequency to the expected frequency of occurrences within each window along this 'suitability axis' (Hirzel et al. 2006). Since the Boyce index is the Pearson correlation coefficient of the predicted to expected ratio of the windows and their mean habitat suitabilities, a sizeable number of evaluation points is necessary to calculate this metric (Breiner et al. 2015, Appendix S1). In ENphylo four different evaluation metrics (AUC, TSS, Boyce index, and Omission Rate) are implemented. Here, we selected the model with highest AUC value over the 50 replicates (this is specified in the text now) albeit we reported also the TSS and Boyce values of the corresponding iteration for

comparison. Although TSS and Boyce are not maximized, the average values of these metrics is comfortably high.

The text now reads:

“Since evaluation metrics can provide misleading results when quantifying the accuracy of models calibrated on sample size lower than 10 presences⁵⁵, we reported the ENphylo performances without considering the models related to the species with less than 10 occurrences. After this filtering, ENphylo achieved excellent predictive performances with an AUC value averaged among the modelled species equal to 0.918 (sd = 0.106), an average TSS equal to 0.703 (sd = 0.166), and an average Boyce index equal to 0.394 (sd = 0.122).”

Hirzel, A. H., Le Lay, G., Helfer, V., Randin, C., & Guisan, A. (2006). Evaluating the ability of habitat suitability models to predict species presences. *Ecological modelling*, 199(2), 142-152.

Breiner, F. T., Guisan, A., Bergamini, A., & Nobis, M. P. (2015). Overcoming limitations of modelling rare species by using ensembles of small models. *Methods in Ecology and Evolution*, 6(10), 1210-1218.

L183-185: Which R packages did you use and, more importantly, what were your model parameters?

RE: We enriched the description of model calibration by adding information about the tuning parameters and the R package used to train the SDMs.

The new version of the main text now reads:

“To model species climatic niche, we adopted the ESM approach for species reporting a number of occurrences between 15 and 30. ESMs were trained by considering all possible combinations of the environmental variables taken two at a time. Lastly, for species with >30 occurrences were modelled by applying a traditional SDM approach (i.e. including all environmental variables at the same time). For both ESMs and SDMs, we adopted an ensemble forecasting approach by testing three widely used modelling techniques: Maximum Entropy (MaxEnt⁴⁹), Random Forest (RF⁵⁰), and Generalized Linear Models (GLM). Models were trained by relying on the functionalities provided in the ‘biomod2’ R package⁵¹. Specifically, we set “quadratic” and `interaction.level = 1` for defining the GLM parameters. For the other algorithms, we maintained the default parameter settings adopted by biomod2.”

In your discussion, I suggest you discuss the assumptions/limitations of your models, especially regarding your rarest species. Do you think that with fewer than 5 occurrences you can capture the whole niche of the species (see van Proosdij et al., 2016 [<https://doi.org/10.1111/ecog.01509>] or Erickson & Smith, 2023 [<https://doi.org/10.1111/ecog.06500>])? You assume that all your species will conserve their niche in the future (especially for ENphylo, where all the links should stay the same). Do you know if local adaptations are possible regarding your species?

RE: We feel it is important to realize that ENphylo *does not* capture the species niche from the datapoints, but from its phylogenetic position. With ENphylo, one

gets a niche even without any occurrence at all, since the niche dimensionality (marginality and specialization) are inferred from the tree and data (niche dimensions) of the sister species. The datapoint use attains to the projection of the niche on the “background” climate via Mahalanobis distance. It is that projection that works with at least 5 datapoints. Having said that, the referee suggestion to discuss the assumptions/limitations of our approach is very much welcome. The main text now reads:

“Predicting species distribution for extremely rare species cannot usually be done^{26,62}, and they are usually omitted¹⁸ or presence data points simulated within the extent of occurrence so that up to 10 geographic cells are included⁶¹. With SDM validation, statistical issues introduced by rarity are still present, as with rare species common validation metrics such the AUC suffers from variation in the species prevalence and sample size^{55,63}. In addition, pseudo-presences necessarily introduce some autocorrelation in the data, and can only be supposed to may harbor the species presence. However, it is important to notice that our procedure is not meant to ‘learn’ the niche from the ‘pseudo-presences’. In ENphylo, the niche of a rare species is imputed after modelling the niches of its relatives with ENFA. True presences and ‘pseudo-presences’ are just used to convert the imputed niche into habitat suitability, by computing the Mahalanobis distance between sampled and the ‘mean’ habitat^{39,64}. Since the ‘pseudo-presences’ are sampled exactly according to their similarity to the real presence datapoints, this distance could be safely assumed to be left almost unaltered by our procedure. Of course, this does not mean that five datapoints provide a faithful representation of the species niche, especially because we selected the pseudo-presences as to minimize their dissimilarity from the true presences. This may possibly imply the rare species niches are probably narrow. We believe maintaining this non-random narrowness, though, provides a more genuine representation of the actual niche than producing an artificially broad niche for an otherwise little-known species. As a matter of fact, ENphylo predictive performance was quite high (AUC > 0.9) which testifies that the modelling procedure was robust. By applying ENphylo, and by testing explicitly the combined effects of land-use and climate change under different scenarios, we found that the most serious threat to Madagascar’s future chameleon biodiversity is posed by land conversion to agriculture and other human activities, rather than by climate change.”

We are clear ENphylo cannot predict the species niche in the future. Yet, the niche dimensions attributed to the rare species are inferred from varying topologies and branch lengths, which means potential limits in the inferred distribution of the niche axes can be derived from the simulations. This is a potentially fruitful area for investigation in the future, we are grateful to the reviewer for pointing this out.

Literature cited

18: Wiens, J. J. & Zelinka, J. How many species will Earth lose to climate change? *Glob Chang Biol* 30, e17125 (2024).

26: Lomba, A. et al. Overcoming the rare species modelling paradox: A novel hierarchical framework applied to an Iberian endemic plant. *Biol Conserv* 143, 2647–2657 (2010).

62: Erickson, K. D. & Smith, A. B. Modeling the rarest of the rare: a comparison between multi-species distribution models, ensembles of small models, and single-species models at extremely low sample sizes. *Ecography* 2023, e06500 (2023).

63: van Proosdij, A. S. J., Sosef, M. S. M., Wieringa, J. J. & Raes, N. Minimum required number of specimen records to develop accurate species distribution models. *Ecography* 39, 542–552 (2016).

64: Clark, J. D., Dunn, J. E. & Smith, K. G. A Multivariate Model of Female Black Bear Habitat Use for a Geographic Information System. *J Wildl Manage* 57, 519 (1993).

REVIEWERS' COMMENTS:

Reviewer #1 (Remarks to the Author):

Well done!

Reviewer #2 (Remarks to the Author):

The authors did a good job in answering all my comments and review the manuscript. I have no more comments on this interesting paper.